# Prediction of the Effects of Missense Mutations on Human Myeloperoxidase Protein Stability Using In Silico Saturation Mutagenesis

**DOI:** 10.3390/genes13081412

**Published:** 2022-08-08

**Authors:** Adebiyi Sobitan, William Edwards, Md Shah Jalal, Ayanfeoluwa Kolawole, Hemayet Ullah, Atanu Duttaroy, Jiang Li, Shaolei Teng

**Affiliations:** 1Department of Biology, Howard University, Washington, DC 20059, USA; 2Department of Electrical Engineering and Computer Science, Howard University, Washington, DC 20059, USA

**Keywords:** myeloperoxidase (MPO), in silico saturation mutagenesis, missense mutations, post-translational modification (PTM) sites, protein stability

## Abstract

Myeloperoxidase (MPO) is a heme peroxidase with microbicidal properties. MPO plays a role in the host’s innate immunity by producing reactive oxygen species inside the cell against foreign organisms. However, there is little functional evidence linking missense mutations to human diseases. We utilized in silico saturation mutagenesis to generate and analyze the effects of 10,811 potential missense mutations on MPO stability. Our results showed that ~71% of the potential missense mutations destabilize MPO, and ~8% stabilize the MPO protein. We showed that G402W, G402Y, G361W, G402F, and G655Y would have the highest destabilizing effect on MPO. Meanwhile, D264L, G501M, D264H, D264M, and G501L have the highest stabilization effect on the MPO protein. Our computational tool prediction showed the destabilizing effects in 13 out of 14 MPO missense mutations that cause diseases in humans. We also analyzed putative post-translational modification (PTM) sites on the MPO protein and mapped the PTM sites to disease-associated missense mutations for further analysis. Our analysis showed that R327H associated with frontotemporal dementia and R548W causing generalized pustular psoriasis are near these PTM sites. Our results will aid further research into MPO as a biomarker for human complex diseases and a candidate for drug target discovery.

## 1. Introduction

Myeloperoxidase (MPO) is a member of the human peroxidase family of proteins expressed mainly in immune cells such as neutrophils. This family of proteins catalyzes the breakdown of peroxides and is involved in killing foreign bodies [1]. Specifically, invading bacteria enhances their production of hydrogen peroxide. MPO uses hydrogen peroxide to produce chloramine and hypochlorite in the peroxidase or halogenation cycle. These chemicals are cytotoxic to invading bacteria. MPO also uses hydrogen peroxidase in the oxidation of chloride ions to hypochlorous acid [2]. Previous studies have shown that MPO has a polycationic nature; therefore, MPO has a high affinity for cell surfaces ranging from immune cells to lipoproteins and epithelial cells [3]. MPO binding to these surfaces can cause functional changes. For example, the binding of MPO to platelets causes the reorganization of the platelet cytoskeleton and thus alters the aggregation properties [3].

Many heritable missense mutations identified from past research are now known to result in human disease phenotypes. These missense mutations cause disease by disrupting the biosynthesis of MPO [4]. Myeloperoxidase deficiency (MPOD) is the most common neutrophil biochemical mutation that manifests in the phenotype as a lack of peroxidase activity [5]. MPOD was considered a rare disease, but recent studies have shown that MPOD has a high incidence rate and prevalence. In the United States and Europe, MPOD affects 1 in 2000–4000 people [6]. Patients that exhibit MPO deficiency have limited ability to attack and kill pathogenic invaders. The R569W mutation leads to malformation of the MPO protein and the inability of neutrophil cells to complete the biosynthetic pathway of the MPO molecule by not inserting a heme into MPO precursors [7], thus stifling posttranslational processing and stopping the production of enzymatically active MPO species [7]. MPO missense mutations like R548W and R590C are associated with skin conditions, the most common being pustular skin disease and psoriasis [8,9]. Still, other identified missense mutations can cause cognitive diseases—for example, R327H in individuals with frontotemporal dementia and C285G and D403E having genetic associations with autism spectrum disorders [10,11,12].

Research in loss-of-function variants and linking these mutations to disease is central toward the future growth of genomic studies—especially those focused on computational analysis. Constructing quality databases using various tools and models of post-translational analysis, genome-wide association studies (GWAS), and targeted computational genomic analyses allow research to treat and possibly prevent future diseases. Unlike SNVs, loss-of-function variants are not well understood, and more research in computational biology is necessary to assist genetic data-driven approaches for application in human health [13].

Respiratory burst within neutrophils mediates the ejection of MPO, which is then catalyzed by hydrogen peroxide (H2O2). Upon the release of hydrogen peroxide, MPO oxidizes chlorine to hypochlorous acid (HOCl) [14]. Oxidative phosphorylation leads to the production of reactive oxygen species (ROS), which are products of MPO activities. However, phosphorylation of MPO has not been well-studied. Global analysis and mass spectrometry of MPO derived from neutrophils identified five different types of N-glycans or complex sugars at specific locations [15]. N-glycosylation of MPO at certain residues contributes to its functions. In addition to N-glycosylation sites, recent advancement in MS had led to the identification of the heterogeneity of the glycosylation sites on MPO [16]. 

The identification of post-translational modification (PTM) sites through experiments is a well-established process. In addition, most of the available PTM site data were extracted through experiments. However, the process of identifying PTM sites using experiments is labor-intensive and time-consuming. Experiments are limited by the availability of enzymatic reactions, which also require optimization. A promising alternative is a computational approach, which will reduce the labor and time involved. However, due to the inability to mimic in-vivo conditions, computational approaches are used as a preliminary approach. With the development of more rigorous algorithms, the prediction of PTM sites through a computational approach will be more reliable and accurate. In this study, we used Foldx, a computational tool, to analyze the effect of all possible missense mutations of MPO on its stability. Then, we investigated the functional impact of target mutations altering protein stability.

## 2. Materials and Methods

### 2.1. Sequence Selection and Structure Preparation

We selected the sequence isoform chosen as the canonical sequence, Isoform H17 (identifier: P05164-1) [17]. This entry has 745 residues, and no missing residues were reported in the UniProtKB (www.uniprot.org accessed on 20 July 2021) [18]. For the structure, we used the cryogenic crystal structure of human myeloperoxidase Isoform C (PDB ID: 1CXP) from the RCSB Protein Data Bank (PDB) [19]. This structure was crystallized in a previous study through an X-ray method with a high resolution of 1.8 A [19]. We further modify the PDB ID: 1CXP, which is a heterotetramer, into a homodimer. 

### 2.2. Saturated Computational Mutagenesis and Structure-Based Energy Calculation

We applied a Perl programming language in-house script to substitute each residue on the MPO structure to 19 other residues. We used the list of mutations to perform structure-based energy calculations on Foldx [20]. Foldx is a precise protein tool for predicting protein stability changes upon mutations, and its performance accuracy outperformed other prediction tools [21]. Foldx uses the ‘RepairPdb’ command to allow the protein structure to assume its native state. This optimizes the protein structure for further analysis. For stability analysis, Foldx uses the ‘BuildModel’ command to calculate the difference between the folding energy of the wild-type structure and the mutant structure. We then used the resulting folding energy change (ΔΔG) to categorize the effect of each missense mutation. The mathematical equation for calculating folding energy change (ΔΔG) is:ΔΔG _(stability)_ = ΔG _(folding)_ MUT − ΔG _(folding)_ WT

A positive ΔΔG means the missense mutation destabilizes the protein. A negative ΔΔG means the missense mutation stabilizes the protein. Comparatively, the DDGs derived from Foldx deviate from the experimental ΔΔGs by 0.46 kcal/mol (~0.5 kcal/mol) [22]. In addition, 2.5 kcal/mol was identified as the best cutoff for the corrections between functional changes and ΔΔG [23]. Therefore, we placed the effects of missense mutations on protein stability into five categories, in 0.5 kcal/mol increments. These are highly stabilizing (ΔΔG < −2.5 kcal/mol), moderately stabilizing (−2.5 < ΔΔG < −0.5 kcal/mol), neutral (0.5 < ΔΔG < +0.5 kcal/mol), moderately destabilizing (+0.5 < ΔΔG < 2.5 kcal/mol), and highly destabilizing (ΔΔG > 2.5 kcal/mol).

### 2.3. Sequence-Based Prediction of PTM Sites

We performed the prediction of PTM sites on the sequence of the MPO protein using novel algorithms developed by the CUCKOO Workgroup (www.biocuckoo.org accessed on 20 July 2021). These algorithms have been optimized and improved upon to enhance their PTM site prediction performances. Here, we predicted phosphorylation sites, sumoylation sites, and methylation sites using the default medium threshold. We used the NetOGlyc 4.0 [24] and NetNGlyc 1.0 [25] online servers to predict O-linked glycosylation and N-linked glycosylation sites, respectively. The threshold used for identifying positive sites is 0.5 for both N-linked and O-linked glycosylation site prediction. Finally, we used the MusiteDeep webserver to predict different PTM types using the default threshold (0.5) to minimize false positives [26].

### 2.4. Sequence-Based Pathogenicity Prediction

The tools we used for pathogenicity prediction of each computed missense mutation are the Polymorphism Phenotyping v2 (PolyPhen2) [27], Sorting Intolerant From Tolerant (SIFT) [28], and Screening for non-acceptable polymorphisms (SNAP) [29]. The webservers of PolyPhen2, SIFT, and SNAP accept FASTA sequence of the MPO protein as input. These are robust tools that accept and annotate large volume of data. We used the Batch query for all three webservers to analyze all possible missense mutations.

We used R programming language (https://www.r-project.org/ accessed on 20 July 2021) to perform analysis of variance (ANOVA) among the five ΔΔG categories and PolyPhen2, SIFT, and SNAP scores.

### 2.5. Detection of PTM-SNPs

We used a Perl script to curate the PTM and missense mutation data. We queried the dbNSFP for additional functional annotation of the missense Single Nucleotide Polymorphism (SNP) data. The dbNSFP is a comprehensive database for Non-synonymous SNPs and their Functional predictions in the human genome [30]. The Perl Express version 2.5 is available at http://www.perl.org (accessed on 20 July 2021).

We mapped the position of the missense SNP site and PTM site on the canonical FASTA sequence of the MPO protein to obtain the relative distance between the SNP site and the PTM site. We specified a window length of −7 aa to +7 aa. The + or − sign of the relative distance indicates if the SNP site is downstream or upstream of the PTM site, respectively.

## 3. Results

### 3.1. Analysis of Overall Computed Missense Mutations

Computational mutagenesis of the MPO protein generated 10,811 non-redundant missense mutations from 569 residues. In the experimental structure, C316 was omitted due to a modification [31]. FoldX uses an energy calculation approach to categorize the potential effect of each missense mutation. Of the potential non-redundant mutations, 4221 (~39%) mutations have a highly destabilizing effect (ΔΔG > 2.5 kcal/mol), 3486 (~32%) mutations have a destabilizing effect (0.5 < ΔΔG ≤ 2.5 kcal/mol), 2248 (~21%) mutations have a neutral effect (−0.5 < ΔΔG ≤ 0.5 kcal/mol), 811 (~8%) mutations have a stabilizing effect (−2.5 ΔΔG ≤ −0.5 kcal/mol), and 45 (~0.4%) mutations have a highly stabilizing effect (ΔΔG < −2.5 kcal/mol) on the MPO protein (Figure 1). Furthermore, our analysis shows that missense mutations have more highly destabilizing effects within the Myeloperoxidase_like domain (41%) than outside the Myeloperoxidase_like domain (33%).

### 3.2. Top Missense Mutations Affecting MPO Stability

The result of the computational mutagenesis shows the magnitude of the folding energy change caused by each missense mutation. The mean ΔΔG values of the folding energy change caused by all possible missense mutations on a residue, and ΔΔG values of alanine substitution were showed in Figure 2A. Figure 2B shows the top folding energy changes caused by individual missense mutations. The residues G402, G361, G655, G595, and G189 have the highest mean ΔΔG values among all MPO amino acids. Meanwhile, the residues D264, D260, G501, G615, and T428 have the lowest mean ΔΔG values in MPO residues. The top missense mutations stabilize and destabilize the MPO protein significantly. As shown in Figure 2B, the overall top destabilizing missense mutations G402W, G402Y, G361W, G402F, and G655Y caused a ΔΔG of 83.11 kcal/mol, 81.0 kcal/mol, 75.1 kcal/mol, 70.33 kcal/mol, and 62.39 kcal/mol, respectively. Interestingly, three of these top destabilizing mutations are on residue G402, and these mutations involve the substitution of the glycine residue. The presence of glycine in these positions must contribute to the stability of the MPO protein. Also shown in Figure 2B, the overall top stabilizing missense mutations D264L, G501M, D264H, D264M, and G501L caused a ΔΔG of −4.82 kcal/mol, −4.25 kcal/mol, −4.18 kcal/mol, −3.84 kcal/mol, and −3.79 kcal/mol, respectively. 

We used PyMol to observe the spatial changes caused by key missense mutations on the MPO structure, as shown in Figure 3. In addition, there is an increase in Van der Waals (steric) strain and clashes in G361W, G402W, and G402Y. G402W had eight possible rotamers with steric strains ranging from 92.84 to 129.27. G402Y has five possible rotamers with steric strains ranging from 102.21 to 129.58. Our Foldx output also shows reduction in the sidechain hydrogen bond, backbone hydrogen bond, and Van der Waals attraction of the missense mutations shown in Figure 3 (Appendix A). 

### 3.3. Sequence-Based Analysis of the Effects of Missense Mutations on MPO Stability

We analyzed the damaging and pathogenic impact of missense mutations on the MPO protein sequence. Figure 4 shows boxplots comparing sequence-based predictions against structure-based predictions. The five groups are statistically different from one another (*p*-value < $2\times 10^{-16}$). We observe that the highly destabilizing (ΔΔG > 2.5) category has the most pathogenic and damaging impact on the MPO protein, while the neutral (−0.5 < ΔΔG ≤ 0.5) category has the least pathogenic and damaging impact.

Table 1 shows the structure-based and sequence-based prediction effects of ten missense mutations on MPO stability. Polyphen2, SNAP, and SIFT predicted the top five highly destabilizing missense mutations, predicted by Foldx, to affect the MPO protein. The SNAP tool predicted that the top five stabilizing missense mutations would affect MPO protein function. On the other hand, SIFT tool predicted D264L and D264H as tolerant missense mutations, while Polyphen2 predicted D264H as benign.

### 3.4. Computational Predictions of Disease Phenotypes

We searched through the HGMD database and found 14 missense mutations associated with the MPO protein as shown in Table 2. Scientists have determined these mutations through experiments to cause disease phenotypes in humans. Foldx predicts all the disease-causing mutations to significantly destabilize the MPO structure except G501S, which increased the folding energy of MPO by only 0.04 kcal/mol. Of the 14 HGMD missense mutations, PolyPhen2 predicts A332V, C285G, and M251T as benign. SIFT predicts D371G and A332V as tolerant mutations. Lastly, SNAP predicts that only A332V does not affect MPO stability.

Noticeably, four distinct complex diseases were associated with missense mutations of the MPO protein. For example, R548W, R569W, R327H, and C285G were implicated in general pustular psoriasis (GPP), MPOD, Frontotemporal dementia (FTD) and autism spectrum disorder (ASD), respectively. Based on our computational predictions, R548W, R569W, R327H, and C285G increased the folding energy of MPO by >3 kcal/mol, thereby destabilizing the MPO structure.

Figure 5 highlights the top destabilizing missense mutations associated with the four complex diseases. Arginine (R) substitution with Tryptophan (W) in positions 548 and 569 alters the local environment by constricting neighboring residues. The change in orientation of the sidechains of C285 and R327 due to substitution with Glycine (G) and Histidine (H), respectively, alters their local environment leading to instability of the MPO protein.

### 3.5. Effect of Missense Mutations on Post-Translational Modification Sites on MPO

We used MusiteDeep predictions for our final analysis. MusiteDeep predicted 14 PTM sites within residues 167–745, five phosphorylation sites, two ubiquitination sites, one pyrrolidone carboxylic acid site, five N-Glycosylation sites, and one acetylation site. Figure 6 shows that S231 has the highest mean folding energy change (ΔΔG), 16.23 kcal/mol, while N729 has the least mean ΔΔG, −0.43 kcal/mol. All possible predicted missense mutations on S231 increase the folding energy of the wildtype MPO protein by at least 0.83 kcal/mol. The mutation S231Y increases ΔG by 49.81 kcal/mol, and N729P decreases ΔG by −1.94 kcal/mol. The ubiquitination site K556 shows the second-highest mean ΔΔG of 3.97 kcal/mol. Of 400 missense mutations predicted on 14 PTM sites on MPO, S231Y and N729P produce the highest and lowest ΔΔG, respectively.

Among the 14 missense mutations in MPO causing disease phenotypes, two are close to PTM sites (Table 3). R327H causes frontotemporal dementia (FTD) in humans, and it is located 4-position downstream (+4) to an N-lined glycosylation site, N323. R548W causes generalized pustular psoriasis (GPP) in humans, and it is located 5-position upstream (−5) to T553, a phosphorylation site.

## 4. Discussion

MPO is present in unicellular to multicellular organisms. A well conserved myeloperoxidase_like domain in the human MPO highlights its importance, and the presence of a more conserved An_peroxidase domain in different organisms offers insight into its essential function. MPO plays a pivotal role in the host’s innate and adaptive immunity [32]. MPO causes inflammation when it extravasates into the extracellular environment in high amounts [33]. However, MPO has become a therapeutic target. Some studies found MPOD as beneficial in preventing myocardial infarction [34]. It has been established through experiments that most missense mutations result in a less stable protein [35]. Our study explores the use of computational tools to predict the effect of missense mutations on MPO stability. Protein stability critically affects its function. Based on our predictions, ~71% of all possible mutations will destabilize the MPO protein compared to ~8% stabilizing the MPO protein. Missense mutations increasing protein stability could also alter its function, which could drive disease mechanisms or the evolution of new functions [36]. 

Experimental assays used in the characterization of disease-associated missense mutations exhibit selective sensitivity [37]. However, computational approaches provide unbiased insight into the functional characterization of disease-associated missense mutations [38,39]. Our structure-based analysis provided the basis for understanding how missense mutations could alter the structure of MPO and ultimately affect its function. In Figure 3, we showed that missense mutations that increase steric clashes would destabilize the structure of MPO. We highlighted the top ten mutations significantly affecting the MPO structure (Table 1). Furthermore, we observed that missense mutations that significantly destabilize or stabilize MPO protein could have damaging effects on its function. G402 and G361 contribute to the hydrophobic core of the MPO structure. Therefore, the substitution of glycine with tryptophan (G402W, G361W) or tyrosine (G402Y) in these positions caused steric clashes preventing the normal folding process of the active MPO protein. The top five most stabilizing missense mutations were in D264 and G501. Substituting aspartic acid with leucine (D264L), methionine (D264M), or histidine (D264H) must have a greater substantial hydrophobic effect on the MPO structure. Similarly, G501M and G501L significantly increase MPO stability. We also used sequence-based tools, Polyphen2, SIFT, and SNAP, for mutation pathogenicity prediction. We showed that the top destabilizing/stabilizing mutations have damaging effects on protein function, except for D264H, which is predicted to have neutral effects (Table 1). In our previous study [40], we discovered these tools are reliable for analyzing the effects of mutations on protein function.

A previous study utilized Foldx for mutagenesis to identify deleterious missense mutations [38]. In our research, Foldx predictions of disease-associated missense mutations of MPO showed destabilizing effects except for G501S. The whole-exome sequencing of families having a child with Autistic disorder (ASD) reveals the contributions of de novo (DN) missense mutations, C285G, to ASD diagnosis in affected probands [41]. Another study analyzed de novo mutations in Whole-exome sequencing (WES) data from 5947 families and found that D403E is a genetic risk of ASD [42]. A study discovered that Y173C contributed to the degradation of MPO in the cell. MPO degrades because Y173C prevents its proteolytic maturation and secretion, leading to MPOD [43]. The biochemical characterization of hereditary MPOD in a father and daughter shows alteration (14-base deletion) in their MPO mRNAs. The presence of M251T in some patients led to late degradation of the MPO protein after an early granulocyte differentiation [44]. A study found R327H in the MPO protein after a comprehensive investigation in 38 patients with Dementia. In addition, patients with R327H exhibit behavioral changes associated with Alzheimer’s disease (AD) [45]. Bone marrow analyses of subjects in Italy with MPOD reveal the structural changes of the MPO protein by A332V, D371G, L572W, and W643R [46]. In two independent studies among Japanese patients, scientists discovered two novel missense mutations, R499C and G501S, associated with MPOD. Although rare, R499C and G501S are proximal to H502, a functionally important residue for iron binding. Further analysis revealed that R499C and G501S disrupt the maturation of the MPO protein [47,48]. Nauseef and others discovered R569W in patients with complete MPOD. Restriction digestion of the gDNA of patients with complete MPOD shows a substitution at exon 10, translating to a substitution at codon 569 [7]. A study discovered R548W and R590C in the blood samples of patients with general pustular psoriasis (GPP). R548W and R590C also completely depleted MPO activity in these patients with GPP [9]. From Table 1, we found these missense mutations to destabilize the MPO protein structure. This structural destabilization might play a role in the disease phenotypes discovered in the different patients.

Polyphen2 predicted all disease-associated missense mutations to be probably damaging except A332V, C285G, and M251T. SIFT scores showed all to be deleterious except D371G and A332V. Lastly, the SNAP scores showed all to affect MPO protein but A332V. While Polyphen2, SIFT, and SNAP predicted A332V as not damaging or deleterious, Foldx predicted A332V as destabilizing. In a previous study, the expression of A332V presented an asymptomatic phenotype with partial MPO activity. SIFT, Polyphen2, and SNAP tools rely on the characterization of known protein sequences in databases and are limited in predicting the damaging effects of missense mutations. However, Foldx utilizes the structural information for predicting the protein stability changes upon mutations. 

Whether MPO undergoes other post-translational modifications is not well known, However, MPO undergoes N-linked glycosylation [14]. With the deep learning framework, MusiteDeep, we predicted five different types of post-translational modifications on the MPO protein. We analyzed the effects of all possible missense mutations on the predicted PTM sites and found that mutation of the phosphorylation site, S231, has the most significant impact on destabilizing the MPO protein. T553, another phosphorylation site, could play an important role in modifying MPO. A study demonstrated that the catalytic activity of MPO is regulated through phosphorylation [49]. The substitution of asparagine with proline in all five N-glycosylated sites had significant impacts on the folding energy of MPO. Glycosylation increases the catalytic activity of MPO, and substitution with proline could lead to a decrease or absence of catalytic activity. The ubiquitination site, K556, showed a significant increase in folding energy when mutated. All possible missense mutations at K556 increased the folding energy of the wild-type MPO protein. The mutation of K556 might lead to abnormal or absence of MPO degradation. MusiteDeep also predicted that an acetylation site, K431. Our result showed that mutation of K431 would destabilize the MPO protein by changing its overall charge.

Missense mutations can disrupt PTM sites, thereby leading to diseases [50]. We provided additional analysis on how missense mutations can affect PTM. We assumed that the location of a missense mutation relative to the PTM site would influence the outcome of PTM. We mapped disease-associated missense mutations with our predicted PTMs, and we found two disease-associated PTM-SNPs, which indicates a correlation. The inhibition of T553 and N323 on MPO could influence the progression of generalized pustular psoriasis and frontotemporal dementia (FTD), respectively.

## 5. Conclusions

Based on our predictions, over 70% of all possible mutations will destabilize the MPO protein compared to 8% stabilizing missense mutations. The residues, G402 and G361, contribute to the hydrophobic core of the MPO structure, and their mutations destabilize MPO. Conversely, mutations of D264 and G501 significantly increase MPO stability. Computational analysis of disease-associated missense mutations provides compelling evidence that Foldx is a reliable tool. Computational prediction of missense mutations on PTM sites on MPO revealed the vital role of the different PTM types on the MPO protein. Furthermore, missense mutations located close to PTM sites T553 and N323 on MPO could influence the progression of generalized pustular psoriasis and frontotemporal dementia (FTD), respectively. However, in-vitro analysis of these computational missense mutations is required.

## Figures and Tables

**Figure 1 genes-13-01412-f001:**
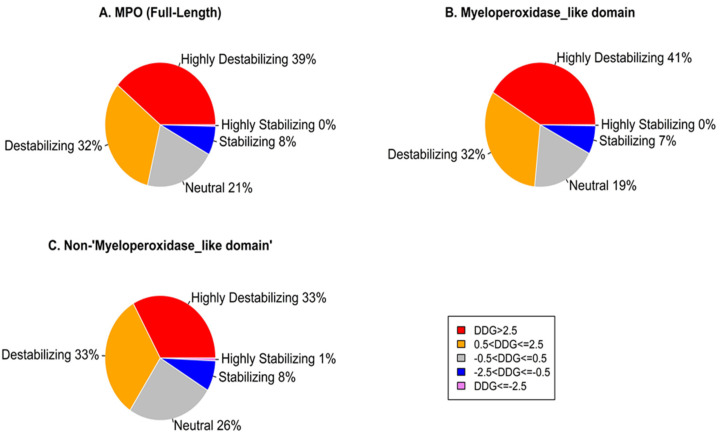
Distribution of the effects of missense mutations on MPO stability. (**A**) residues 167−744 of the whole sequence of 1CXP.pdb; (**B**) residues 317−728 of the Myeloperoxidase_like domain; (**C**) residues 167−315 & 729−744 outside the Myeloperoxidase_like domain.

**Figure 2 genes-13-01412-f002:**
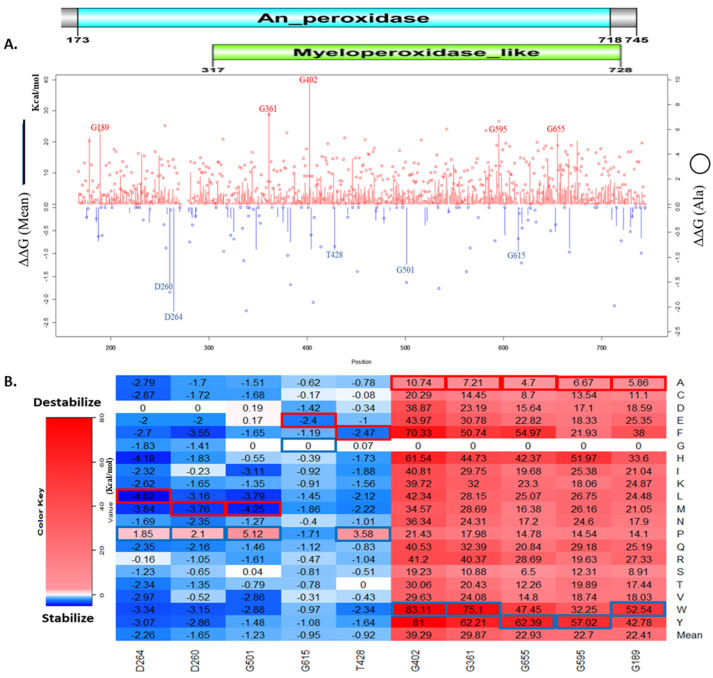
Effects of missense mutations on MPO stability. (**A**) line chart of residues 167−744 showing their mean folding energy changes (ΔΔG) in vertical bars and ΔΔG of alanine substitutions in circles. The An_peroxidase (Aqua bar) and the Myeloperoxidase (green bar) domains represent the superfamily and family domains, respectively; (**B**) heatmap of top 5 stabilizing and top 5 destabilizing mean ΔΔG. The blue rectangle represents maximum values and the red rectangle represents minimum values for each residue position. Key residues are highlighted on the line chart and heatmap.

**Figure 3 genes-13-01412-f003:**
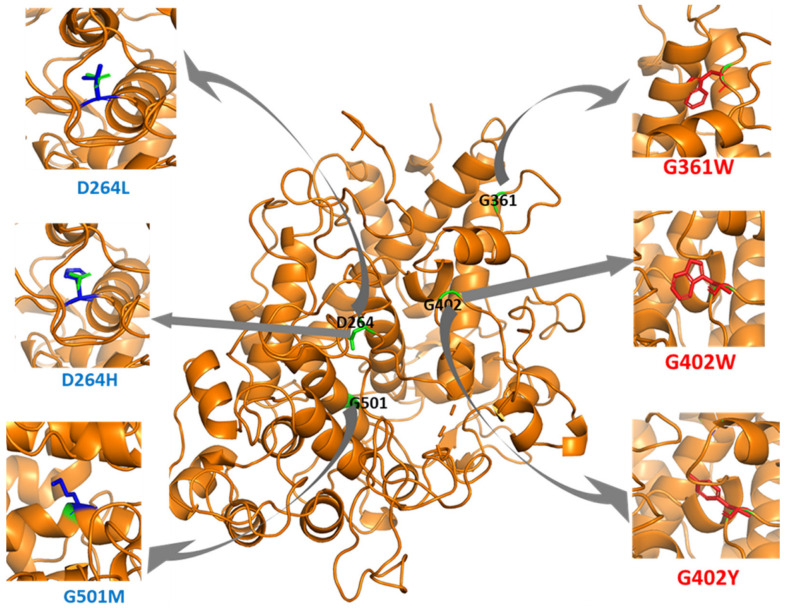
Structural representation of some key missense mutations. Missense mutations in red destabilize the MPO protein (**Right**); Missense mutations in blue stabilize the MPO protein (**Left**); Green represents the wild-type residues.

**Figure 4 genes-13-01412-f004:**
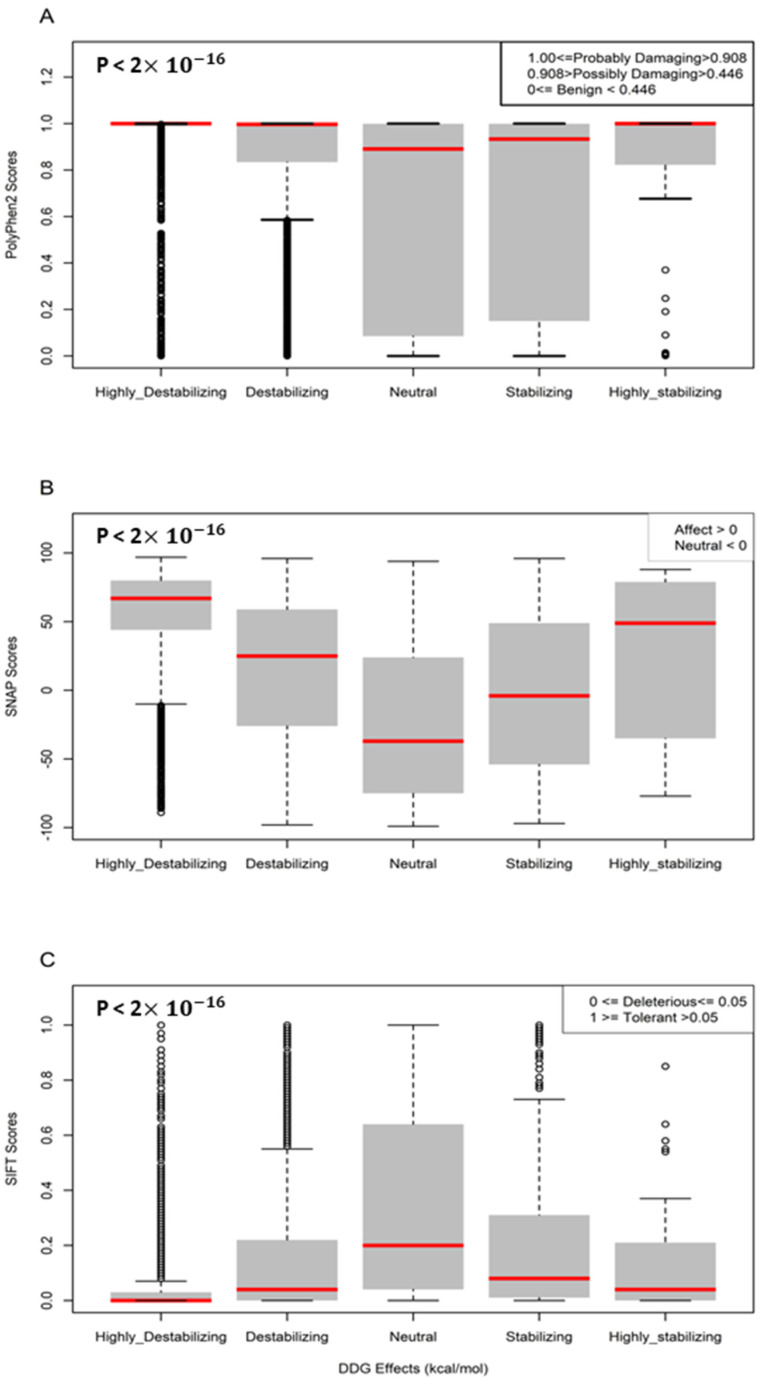
Boxplots of the prediction of mutation pathogenicity. (**A**) PolyPhen2 scores; (**B**) SNAP scores; and (**C**) SIFT scores against five categories of the effects of missense mutations on MPO protein stability.

**Figure 5 genes-13-01412-f005:**
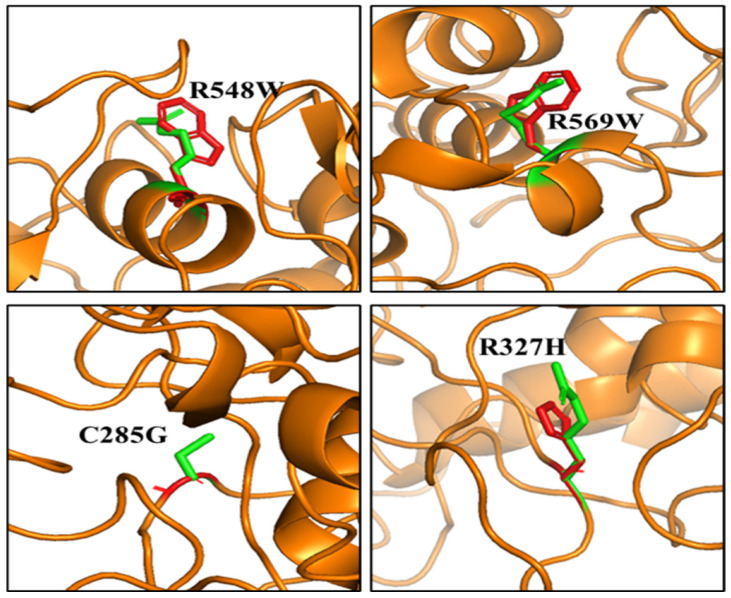
Structural representation of some disease-associated missense mutations of MPO; wildtype residues in Green; mutant residues in Red.

**Figure 6 genes-13-01412-f006:**
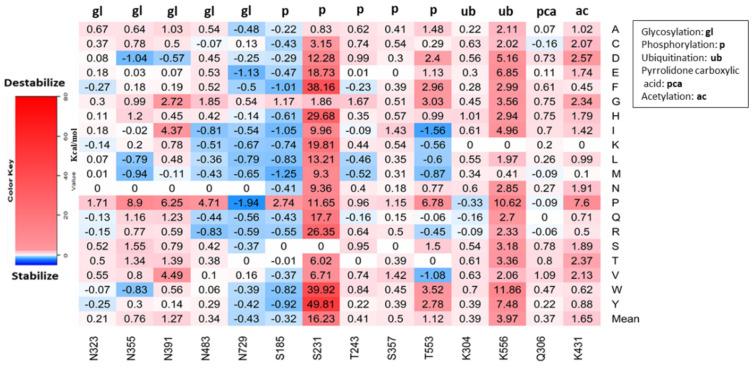
Post−translational modification sites on MPO showing calculated folding energy changes (ΔΔG) for all possible reference residues. Legends are inset.

**Table 1 genes-13-01412-t001:** Functional effects of top ten mutations altering MPO protein stability.

	Structure-Based Prediction (Protein Stability)		Sequence-Based Prediction (Mutation Pathogenicity)					
	FoldX		PolyPhen2		SIFT		SNAP	
Missense Mutations	ΔΔG (Kcal/mol)	Mutation Effect	Score	Effect	Score	Effect	Score	Effect
G402W	83.11	H_Des	1	Pro_d	0	Del	80	Aff
G402Y	81	H_Des	1	Pro_d	0	Del	75	Aff
G361W	75.1	H_Des	1	Pro_d	0	Del	93	Aff
G402F	70.33	H_Des	1	Pro_d	0	Del	77	Aff
G655Y	62.39	H_Des	1	Pro_d	0.01	Del	84	Aff
D264L	−4.82	H_Sta	0.972	Pro_d	0.11	Tol	20	Aff
G501M	−4.25	H_Sta	1	Pro_d	0	Del	85	Aff
D264H	−4.18	H_Sta	0.191	B	0.09	Tol	11	Aff
D264M	−3.84	H_Sta	0.995	Pro_d	0.05	Del	44	Aff
G501L	−3.79	H_Sta	1	Pro_d	0	Del	84	Aff

*Note*: H_Des = Highly Destabilizing, H_Sta = Highly Stabilizing. Pro_d = Probably damaging, B = Benign, Del = Deleterious, Tol = Tolerated, Aff = Affect protein.

**Table 2 genes-13-01412-t002:** Sequence-based and structure-based prediction of disease-associated missense mutations.

	HGMD	Structure-Based Prediction (Protein Stability)		Sequence-Based Prediction (Mutation Pathogenicity)					
		FoldX		PolyPhen2		SIFT		SNAP	
Missense Mutations	Phenotype	ΔΔG (Kcal/mol)	Mutation Effect	Score	Damaging Effect	Score	Effect	Score	Effect
R548W	GPP	11.22	H_Des	1	Pro_d	0	Del	84	Aff
R569W	MPOD	7.84	H_Des	1	Pro_d	0.04	Del	71	Aff
W643R	MPOD	5.92	H_Des	1	Pro_d	0	Del	93	Aff
M251T	MPOD	4.81	H_Des	0.032	B	0.01	Del	35	Aff
R327H	FTD	4.72	H_Des	0.986	Pro_d	0	Del	57	Aff
C285G	ASD	3.31	H_Des	0.239	B	0	Del	74	Aff
Y173C	MPOD	3.08	H_Des	0.995	Pro_d	0	Del	71	Aff
D371G	MPOD	2.25	Des	0.999	Pro_d	0.12	Tol	64	Aff
R499C	MPOD	1.62	Des	0.999	Pro_d	0	Del	62	Aff
R590C	GPP	1.32	Des	1	Pro_d	0	Del	39	Aff
D403E	ASD	1.27	Des	1	Pro_d	0	Del	13	Aff
A332V	MPOD	1.07	Des	0.059	B	0.1	Tol	−26	N
L572W	MPOD	0.51	Des	1	Pro_d	0	Del	73	Aff
G501S	MPOD	0.04	N	0.938	Pro_d	0	Del	23	Aff

*Note:* MPOD = Myeloperoxidase deficiency, FTD = Frontotemporal dementia, ASD = Autism spectrum disorder, GPP = generalized pustular Psoriasis. H_Des = Highly Destabilizing, N = Neutral, Des = Destabilizing. Pro_d = Probably damaging, B = Benign, Del= Deleterious, Tol = Tolerated, Aff = Affect protein.

**Table 3 genes-13-01412-t003:** Disease-associated PTM-SNPs on the MPO protein.

rs_dbSNP151	Missense SNP	PTM SITE	PEPTIDE	DISTANCE APART (PTM ↔ SNP SITE)	PTM TYPE	Phenotype
rs760619802	R327H	N323	GSNITIRNQI	+4	N-LINKED GLYCOSYLATION	FTD
rs148802625	R548W	T553	ILRGLMATPA	−5	PHOSPHORYLATION	GPP

*Note*: SNP represents missense mutation at the protein level, FTD = Frontotemporal dementia, GPP = generalized pustular Psoriasis.

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
