# Peer review of "Prediction of the Effects of Missense Mutations on Human Myeloperoxidase Protein Stability Using In Silico Saturation Mutagenesis"

_genes, 2022, doi:10.3390/genes13081412_

Round 1

Reviewer 1 Report

Missense mutations on MPO are associated with human diseases but their effects on protein functions are understudied. In this manuscript, authors performed comprehensive analysis of the effects of 10,811 potential missense mutations on MPO stability using computational approaches. Top destabilizing mutations have been identified and characterized. Effects of mutations on protein PTM sites have further been analyzed.  Overall, this is a well-written manuscript. Results are clearly present, and conclusions are strongly supported. I only have the following minor comments:

1) In free energy calculation, authors selected Foldx to predict the folding free energy changes upon mutations. Any evidence showing the advantages of Foldx compared to other methods? 

2) For the top destabilizing mutations, the effects of mutations on salt bridges and H-bonds could be analyzed to reveal their effects on protein structures.

3) In the category of ddG effects, how are the cut-offs (+/-0.5 and +/-2.5) determined? Units for free energies are missing in the figures.

Author Response

We thank for reviewer’s valuable comments. The response is shown below.

1) In free energy calculation, authors selected Foldx to predict the folding free energy changes upon mutations. Any evidence showing the advantages of Foldx compared to other methods?

>>Thank you for this comment.

We have added and cited a previous work done to prove the comparative performance of Foldx.

Page 2, line 99-100: “Foldx is a precise tool for predicting protein stability changes upon mutations, and its performance accuracy outperformed other prediction tools [22]”

2) For the top destabilizing mutations, the effects of mutations on salt bridges and H-bonds could be analyzed to reveal their effects on protein structures.

>> Thank you for your suggestion.

Page 5, line 192-194: We have included our outputs for the H-bonds and Van der Waals attraction. We have also affixed a comprehensive output for the top ten missense mutations in Supplementary Table 1.

3) In the category of ddG effects, how are the cut-offs (+/-0.5 and +/-2.5) determined? Units for free energies are missing in the figures.

>> Thank you for this recommendation.

Page 3, line 109-110: We have added how the cut-off was determined and cited the literature. “Comparatively, the ΔΔGs derived from Foldx deviate from the experimental ΔΔGs by 0.46 kcal/mol (~0.5 kcal/mol) [23]. 2.5 kcal/mol was identified as the best cutoff for the corrections between function changes and ΔΔG [24].”

We have also updated our figures with the free energy units.

Reviewer 2 Report

Here Sobitan and colleagues describe the in silico mutagenesis prediction of missense mutations on MPO protein stability. The investigators used the structure-based energy calculation on Foldx and sequence-based pathogenicity prediction programs, PolyPhen2, SIFT, and SNAP. Further, they verified their findings with described phenotypes of MPO mutations.

Overall a very interesting paper, which marks a significant approach for the future. However, I have a couple of comments:

11.      General: I am missing a clear hypothesis. After reading through the paper a couple of times, it is still unclear to me what the main point is here, the different MPO mutations and their influence or the use of Foldx as a good predictor.

22.      Introduction: It would be nice to spell out PTM here once as well as you did with MPO and not just in the abstract.

33.       Methods/Results: There is a difference in the categorization of DDG as in Methods the limits are -2.0 / -0.5 / 0.5 / 2.0 and in the whole results -2.5 / -0.5 / 0.5 / 2.5

44.       Results: Line 164-165: Perhaps include the word “overall” top destabilizing mutations, as it is otherwise a bit confusing compared to the once with highest and lowest means you mentioned above. Same for stabilizing mutations mentioned in line 170.

55.       Figure 2A: It is not clear to me what you want to visualize with the aqua and green bar on top of it. Further, it would be better to change the colour of the legend line and circle as they display in blue, which is the same colour as you choose for negative means.

66.       Figure 2B: I know that you wanted to point out the same directions when you highlight some of the rectangles in blue and red. However, as the heat map has all the different shades of red and blue, some of the highlights are not seen well, e.g. D260P. Further, the whole figure seems a bit blurred, which makes it difficult to read, especially when you print it.

77.       Results: Line 178-182: Please include to reference to Figure 3 here.

88.       Results: Line 190-191: It is a bit hard to believe really all five groups are significantly different when looking at Fig 4A, blot 3 and 4. And how did you calculate this, as it isn’t mentioned in the methods?

99.       Figure 4: For better visualization, I would suggest addressing the different plots, not as e.g. DDG<2.5 but better as their given categories (“highly destabilizing). Further marking the limit in each score with a small horizontal line will increase the visibility.

110.   Results: Section 3.4: You also display the stabilizing mutation; however, you are more indifferent to the different prediction tools, would be nice to assess this a bit more here. Additionally, in line 200, it is not just the destabilizing mutation that affects the protein, as stabilizing also affects a protein. Here would wording like “negatively affects” be more clearly.

111.   Results: Section 3.5: The sentence in line 242 (“The mutation S231Y increases ΔG by 49.81 kcal/mol, and N729P decreases 242 ΔG by -1.94 kcal/mol”) would, in my opinion, fit better after before the line before about the ubiquitination (line 241). Like it is, you jump from examples of phosphorylation to ubiquitination and back to phosphorylation.

112.   Discussion: Line 277: I think the reference to Figure 4 should be Table 1.

113.   Discussion: Lines 284-285: You mentioned that the sequence-based prediction and the structure-based coincide; however, this is not true for some stabilizing mutations, as you mentioned in 201-202, respectively.

114.   Discussion: Lines 286-314: It is nice to connect the in silico prediction with the known diseases. However, this part seems a bit blown up. I don’t think it is necessary to provide all DDGs again. You can perhaps refer to table 2. Additionally, it is unclear what further information this will give, as this part is just a summary of these findings. When refereeing to the next sentences (lines 315 – 319) it doesn’t need this amount of information.

15.   Discussion: Line 315-319: I am missing a discussion here, as you just mention four findings but no further thoughts about it. What clues could be drawn? Is there a paper with these methods investigating other proteins with the same different accuracy to different methods?

16.   Discussion: Line 307 + 308: Probably there is a typing for G501C, which should be G501S, shouldn’t it?

Author Response

Thank you for providing the detailed comments. The response is shown below and revised parts are highlighted.

  1. General: I am missing a clear hypothesis. After reading through the paper a couple of times, it is still unclear to me what the main point is here, the different MPO mutations and their influence or the use of Foldx as a good predictor.

>>Thank you for your comment

Page 2, line 83-86: We have added a hypothetical statement, “In this study, we used Foldx, a computational tool, to analyze the effect of all possible missense mutations of MPO on its stability. Then, we investigated the functional impact of target mutations altering protein stability”, for more clarity.

  1. Introduction: It would be nice to spell out PTM here once as well as you did with MPO and not just in the abstract.

>> Page 2, line 75: We have written it out and placed the acronym in parenthesis “Post-translational modification (PTM)

  1. Methods/Results: There is a difference in the categorization of DDG as in Methods the limits are -2.0 / -0.5 / 0.5 / 2.0 and in the whole results -2.5 / -0.5 / 0.5 / 2.5

>> Thank you for your observation.

Page 3, line 114-117: We have corrected the categorization as -2.5 / -0.5 / 0.5 / 2.5.

  1. Results: Line 164-165: Perhaps include the word “overall” top destabilizing mutations, as it is otherwise a bit confusing compared to the once with highest and lowest means you mentioned above. Same for stabilizing mutations mentioned in line 170.

>> We sincerely appreciate your honest remark.

Page 4, line 175 & 180: We have included “overall” as it makes more sense that way

  1. Figure 2A: It is not clear to me what you want to visualize with the aqua and green bar on top of it. Further, it would be better to change the colour of the legend line and circle as they display in blue, which is the same colour as you choose for negative means.

>> Thank you for your comment.

The aqua and green bar are the superfamily and family domains, respectively. We have added their description to the figure 2A caption.

Page 6, line 183+ 186-187: We have also changed the legend line and circle to a black color.

  1. Figure 2B: I know that you wanted to point out the same directions when you highlight some of the rectangles in blue and red. However, as the heat map has all the different shades of red and blue, some of the highlights are not seen well, e.g. D260P. Further, the whole figure seems a bit blurred, which makes it difficult to read, especially when you print it.

>> Thank you for your observation.           

We improved the resolution of the figure, and this makes the texts more visible.

  1. Results: Line 178-182: Please include to reference to Figure 3 here.

>> Thank you for your suggestion.

Page 5, line 189: We have referenced Figure 3 in the text.

  1. Results: Line 190-191: It is a bit hard to believe really all five groups are significantly different when looking at Fig 4A, blot 3 and 4. And how did you calculate this, as it isn’t mentioned in the methods?

>> Thank you for this comment

Page 3, line 136 – 138: We added this statement to the method section. “We used R programming language (https://www.r-project.org/) to perform analysis of variance (ANOVA) among the five ΔΔG categories and PolyPhen2, SIFT, and SNAP scores.”.

  1. Figure 4: For better visualization, I would suggest addressing the different plots, not as e.g. DDG<2.5 but better as their given categories (“highly destabilizing). Further marking the limit in each score with a small horizontal line will increase the visibility.

>> Thank you for your suggestion.

Page 7, line 208: We have updated Figure 4 per your suggestion. We have also included the p-values in each boxplot.

  1. Results: Section 3.4: You also display the stabilizing mutation; however, you are more indifferent to the different prediction tools, would be nice to assess this a bit more here. Additionally, in line 200, it is not just the destabilizing mutation that affects the protein, as stabilizing also affects a protein. Here would wording like “negatively affects” be more clearly.

>>Thank you for your suggestion

Page 7, line 215 – 216: We have added this statement, “The SNAP tool predicted that the top five stabilizing missense mutations would affect MPO protein function.”

Page 11, line 291-292: We also included this statement in the discussion section, “Further, we observed that missense mutations that significantly destabilize or stabilize MPO protein could have damaging effects on its function”.

  1. Results: Section 3.5: The sentence in line 242 (“The mutation S231Y increases ΔG by 49.81 kcal/mol, and N729P decreases 242 ΔG by -1.94 kcal/mol”) would, in my opinion, fit better after before the line before about the ubiquitination (line 241). Like it is, you jump from examples of phosphorylation to ubiquitination and back to phosphorylation.

>>Thank you for the observation.

Page 10, line 255 – 256: We have reversed the texts for coherence as you well observed.

  1. Discussion: Line 277: I think the reference to Figure 4 should be Table 1.

>>Thank you.

Page 11, Line 291: We have corrected Figure 4 to Table 1.

  1. Discussion: Lines 284-285: You mentioned that the sequence-based prediction and the structure-based coincide; however, this is not true for some stabilizing mutations, as you mentioned in 201-202, respectively.

>> Thank you for your comment. The sequence-based tools are used to predict the mutation pathogenicity, and the results are correlated with the protein stability predictions from Foldx. We have revised the titles for Table 1 & 2. In addition, we have revised on

Page 11, Line 299 - 303:, “We also used sequence-based tools, Polyphen2, SIFT, and SNAP, for mutation pathogenicity prediction. We showed the top destabilizing/stabilizing mutations have damaging effects on protein function, except for D264H is predicted to have neutral effects (Table 1). In our previous study [41], we discovered these tools are reliable for analyzing the effects of mutations on protein function.”.

  1. Discussion: Lines 286-314: It is nice to connect the in silico prediction with the known diseases. However, this part seems a bit blown up. I don’t think it is necessary to provide all DDGs again. You can perhaps refer to table 2. Additionally, it is unclear what further information this will give, as this part is just a summary of these findings. When refereeing to the next sentences (lines 315 – 319) it doesn’t need this amount of information.

>>Thank you for your suggestion.

Page 11, Line 304-330: The paragraphs have been rewritten and we have taken out the ΔΔGs to avoid repetition.

  1. Discussion: Line 315-319: I am missing a discussion here, as you just mention four findings but no further thoughts about it. What clues could be drawn? Is there a paper with these methods investigating other proteins with the same different accuracy to different methods?

>> Thank you for your comment.

Page 11, line 336 – 339, we have added the statement “SIFT, Polyphen2, and SNAP tools rely on the characterization of known protein sequences in databases and are limited in predicting the damaging effects of missense mutations. However, Foldx utilizes the structural information for predicting the protein stability changes upon mutations.”

  1. Discussion: Line 307 + 308: Probably there is a typing for G501C, which should be G501S, shouldn’t it?

>> Thank you for your observation.

Page 11, line 318+319: This has been corrected
